# A New Viscoelasticity Dynamic Fitting Method Applied for Polymeric and Polymer-Based Composite Materials

**DOI:** 10.3390/ma13225213

**Published:** 2020-11-18

**Authors:** Vitor Dacol, Elsa Caetano, João R. Correia

**Affiliations:** 1CONSTRUCT (ViBEST), Faculty of Engineering (FEUP), University of Porto, 4200-465 Porto, Portugal; ecaetano@fe.up.pt; 2CERIS, DECivil, IST, University of Lisbon, 1049-001 Lisbon, Portugal; joao.ramoa.correia@tecnico.ulisboa.pt

**Keywords:** viscoelasticity, creep and relaxation, interconversion, dynamic behaviour, dynamic mechanical analysis, storage modulus, loss factor

## Abstract

The accurate analysis of the behaviour of a polymeric composite structure, including the determination of its deformation over time and also the evaluation of its dynamic behaviour under service conditions, demands the characterisation of the viscoelastic properties of the constituent materials. Linear viscoelastic materials should be experimentally characterised under (i) constant static load and/or (ii) harmonic load. In the first load case, the viscoelastic behaviour is characterised through the creep compliance or the relaxation modulus. In the second load case, the viscoelastic behaviour is characterised by the complex modulus, $E^{*}$, and the loss factor, η. In the present paper, a powerful and simple implementing technique is proposed for the processing and analysis of dynamic mechanical data. The idea is to obtain the dynamic moduli expressions from the Exponential-Power Law Method (EPL) of the creep compliance and the relaxation modulus functions, by applying the Carson and Laplace transform functions and their relationship to the Fourier transform, and the Theorem of Moivre. Reciprocally, once the complex moduli have been obtained from a dynamic test, it becomes advantageous to use mathematical interconversion techniques to obtain the time-domain function of the relaxation modulus, E(t), and the creep compliance, D(t). This paper demonstrates the advantages of the EPL method, namely its simplicity and straightforwardness in performing the desirable interconversion between quasi-static and dynamic behaviour of polymeric and polymer-composite materials. The EPL approximate interconversion scheme to convert the measured creep compliance to relaxation modulus is derived to obtain the complex moduli. Finally, the EPL Method is successfully assessed using experimental data from the literature.

## 1. Introduction

Polymeric and polymer-based composite materials present viscoelastic behaviour, which is associated with the ability to simultaneously store and dissipate energy when subjected to a mechanical load. This coupled system (elasticity and viscosity) provides an increased rate of damping.

Understanding the response of a polymer or polymer-composite material to dynamic excitation, which is very relevant for many structural applications, involves characterising the physical factors affecting deformation and recovery cycles.

One of the relevant mechanical properties of a polymer is that the modulus, i.e., the ratio of stress to strain, has a complex nature: the real or in-phase part represents the part of the response in phase with the excitation; the imaginary part is associated with the lagging part of the response and expresses the damping capacity of the material.

Furthermore, the interrelationships between linear viscoelastic material functions enable the conversion of certain material functions into other functions that can be more easily obtained over a wide-enough range of time or frequency [1]. An interconversion procedure between time and frequency domain functions can therefore be used in analyses that require the relaxation or compliance time spectrum, E(t) or D(t), instead of the readily available dynamic moduli, $E^{*}\left(\omega \right)$ or $D^{*}\left(\omega \right)$ [2].

A variety of methods exist which are applicable to the interconversion of static (creep) and dynamic (relaxation) functions on the basis of appropriate experimental data of various polymers. Reference [3] presents a method based on the convolution formula of creep compliance to obtain the dynamic modulus using a numerical integration method to solve these integrals. Unfortunately, this method needs to use numerical integration methods to solve complicated integrals and requires complex computing programs. References [4] and [5] present a numerical method of interconversion between linear viscoelastic material functions based on a Prony series representation. Here, the effects of different choices of relaxation and retardation times on the accuracy of the method are discussed. Prony series representation requires the adaptation of many variables; another inherent disadvantage of this approach is the possibility of generating negative coefficients of the Prony series or oscillations, namely when the source data exhibit significant variability. The problem of negative Prony series coefficients is addressed in [6], where the difficulty of the direct fitting of a Prony series function to experimental data without appropriate pre-smoothing is discussed. In Ref. [7], a power-law function is used to express the creep compliance, which is converted into complex compliance.

Despite certain advantages of each of the described methods, all are inefficient to express the complete viscoelastic curve of a polymer or a polymer-based composite, i.e., of covering all stages of their viscoelastic behaviour, whether in creep or relaxation. The EPL function, in turn, is efficient in adjusting experimental creep and relaxation data, and is able to predict viscoelastic behaviour for long periods [1].

When studying the mechanical behaviour of viscoelastic materials, it is usually assumed that the strains and the stresses are deterministic functions. However, Wang et al. [8] highlight the importance of considering the uncertainty propagation of the frequency response for the robust design of viscoelastic damping structures. According to De Lima et al. [9], among the various methods devised for uncertainty modelling, the stochastic finite element method has received major attention, as it is well adapted for applications to complex engineering systems. 

In this paper, an interconversion procedure is proposed to convert the measured complex moduli to a relaxation viscoelastic function, and the measured relaxation modulus to complex moduli. Despite the potential importance, the experimental quantification of uncertainties and the corresponding stochastic modelling are not addressed.

The procedure proposed combines (i) Carson and Laplace transform functions, (ii) the Theorem of Moivre, and (iii) Gamma-function approximation. The proposed interconversion function is easy to implement and leads to optimal curve fitting results in both the time- and frequency-domain, without the requirement of rheological models to represent the polymeric system.

## 2. Theoretical Background

### 2.1. The Hysteresis Basics

The hysteresis cycle can be analytically characterised by assuming that the response to a harmonic excitation will be harmonically similar. In turn, when polymer and polymer-based composite materials are subjected to harmonic stress cycles, they have harmonic deformation cycles with the same frequency but with a difference of phase with respect to the corresponding stress cycles. Reference [10] quotes that, within limits, the plots of measured stress versus measured strain are elliptical in shape and retain that shape as amplitude increases. Furthermore, the slope of the major axis of the ellipse is a measure of the stiffness of the sample, and the ratio between minor and major axis lengths is a measure of the damping. It should be noted that all types of internal damping, and particularly those inherent to the movement of polymer chains, are related to hysteresis cycles. 

From the relationship between strain and stress during a hysteresis cycle of a viscoelastic material, the following relation is obtained
(1)$$\sigma \left(t\right)=\frac{\sigma_{0}}{\varepsilon_{o}}\cos\left(\delta \right)\varepsilon \left(t\right)+\frac{\sigma_{0}}{\varepsilon_{o}}\sin\left(\delta \right)\frac{d\varepsilon \left(t\right)}{dt}$$
where σ(t) is the applied stress; $\sigma_{0}$ is the amplitude of stress during the hysteresis cycle; $\varepsilon_{0}$ is the amplitude of the strain during the hysteresis cycle δ is the angle between excitation and response ε(t) is the strain at time t-_th_.

The first term of Expression (1) represents the storage modulus, $E^{\prime }$, and the second term represents the damping capacity of the viscoelastic material.

Expression (1) can be written as
(2)$$\sigma \left(t\right)=E^{\prime }\varepsilon \left(t\right)+\frac{E^{\prime }}{\omega }\dot{\varepsilon }\tan\left(\delta \right)$$
where $\dot{\varepsilon }$ is the strain rate; $E^{\prime }$ is the storage modulus; ω is the angular frequency; tan(δ) is the so-called “tan-delta”, representing the loss factor η (η=tan(δ)).

### 2.2. A Brief Review of the Complex Moduli

The effect of polymeric material on the damping of a given structure is influenced by the stiffness of the material as well as its damping. In fact, the viscous part of the viscoelastic behaviour of polymers becomes evident under dynamic loading.

By imposing a harmonic excitation of type $\varepsilon \left(t\right)=\varepsilon_{0}\cdot e^{i\left(\omega t\right)}$ on a viscoelastic material, the stress response in the frequency domain, derived from Expression (2), assumes the following form
(3)$$\sigma \left(\omega \right)=E^{\prime }\left(\omega \right)\varepsilon \left(\omega \right)+iE^{\prime }\left(\omega \right)\eta \left(\omega \right)\varepsilon \left(\omega \right)$$
where i is the imaginary part of a complex function given by $\sqrt{-1}$.

In Expression (3), the product given by $E^{\prime }\left(\omega \right)\eta \left(\omega \right)$ is named the loss modulus, $E^{\prime \prime }\left(\omega \right)$, and represents the imaginary part of the complex modulus, $E^{*}\left(\omega \right)$, which is given by
(4)$$E^{*}\left(\omega \right)=E^{\prime }\left(\omega \right)+iE^{\prime \prime }\left(\omega \right)$$

From the complex modulus, the dynamic modulus amplitude, $\left|E^{*}\right|$, is determined as
(5)$$\left|E^{*}\right|=\sqrt{{\left(E^{\prime }\right)}^{2}+{\left(E^{\prime \prime }\right)}^{2}}$$

The storage and the loss moduli represent the storage and the dissipation of energy, respectively. In this respect, as the stress and the internal damping of a viscoelastic material depend on the frequency of the imposed load, the relationship between the dissipated energy, H, and the loss modulus, $E^{\prime \prime }$, may be described as
(6)$$H=\overset{2\pi /\omega }{\underset{0}{\int }}\varepsilon \left(\omega \right)\left[E^{\prime }\left(\omega \right)+iE^{\prime \prime }\left(\omega \right)\right]\frac{\partial \varepsilon }{\partial t}dt=\varepsilon_{0}^{2}\omega \pi E^{\prime \prime }$$

### 2.3. Relationship Between Time and Frequency Domain Moduli 

Considering the relationships defined in the previous section, it can be concluded that the knowledge of the relaxation modulus or the creep compliance is sufficient to describe the behaviour of materials with linear viscoelastic behaviour [7]. Consequently, the complex modulus, $E^{*},\;\mathrm{and}$ therefore its components ($E^{\prime },\;E^{\prime \prime },\;\left|E^{*}\right|\;$ and η), can be defined from the frequency characterisation of the strain and stress. Moreover, the relaxation modulus (or creep compliance) can be obtained from the complex relaxation modulus (or complex compliance).

The relationship between the time and the frequency domain moduli is related to the operational functions given as
(7)$$E^{*}\left(\omega \right)={\tilde{E}\left(s\right)|}_{s=i\omega }$$
where $\tilde{E}$ is the Carson Transform of a function, given by $\tilde{E}\left(s\right)=s\bar{E}\left(s\right)$; and $\bar{E\;}$ is the rational functions of s in Laplace domain.

Moreover, the storage and the loss modulus can be readily deduced from the relaxation modulus via Fourier transform [11]:(8)$$E^{\prime }\left(\omega \right)=\omega \int_{0}^{\infty }E\left(t\right)\sin\left(\omega t\right)dt$$
(9)$$E^{\prime \prime }\left(\omega \right)=\omega \int_{0}^{\infty }E\left(t\right)\cos\left(\omega t\right)dt$$

Consequently, the relaxation modulus (or creep compliance) can be deduced from the complex modulus by the Fourier transform:(10)$$E\left(t\right)=\frac{2}{\pi }\int_{0}^{\infty }\frac{E^{\prime }\left(\omega \right)}{\omega }\sin\left(\omega t\right)d\omega$$
(11)$$E\left(t\right)=\frac{2}{\pi }\int_{0}^{\infty }\frac{E^{\prime \prime }\left(\omega \right)}{\omega }\cos\left(\omega t\right)d\omega$$

However, the Fourier transform of the creep compliance, D(t), is not convergent because it is an increasing function of time. 

Given the above-defined relation (Expression (7)), the following procedure can be followed to derive the complex modulus:(i)From a given creep test data, to determine the creep compliance;(ii)To apply the interconversion method to obtain the relaxation modulus from creep compliance;(iii)To apply the interconversion method to obtain the complex modulus from the relaxation modulus function;(iv)According to the required frequency, to calculate the viscoelastic parameters of the frequency ($E^{\prime }$, $E^{\prime \prime }$, η).

The steps (i) and (ii) involve the concept of the Exponential-Power Law (EPL) method, which is derived from Ref. [12] and presented in Ref. [1]. A brief description of the method is presented in the next section.

## 3. Exponential-Power Law 

In Ref. [13], the similarity between creep behaviour and mechanical system ageing is shown, where the failure rate of mechanical repairable systems that deteriorate over time due to ageing can usually be visualised by a bathtub curve, as shown by the creep rate curve.

In Ref. [12] a differential equation is proposed to synthesise the behaviour of the creep rate expressed by the bathtub curve. Once integrated, this equation yields
(12)$$\varepsilon \left(t\right)=k\cdot t^{\beta }\cdot e^{\frac{t}{t_{i}}}$$
where ε(t) is the strain; β is a constant related to strain hardening *(* β<1 and $1/t_{i}\ll \beta$
*);*
k is a constant related to the main stress influence; and $t_{i}$ is the instability time.

The instability time is the period of time (after the material is loaded) after which there is a risk that the system starts to fail and occurs in tertiary creep stage.

The conversion between the creep compliance, fitted with *Expression (12)), and the relaxation modulus may be required to estimate the fatigue resistance of composite structures. Thus, a numerical interconversion between the creep compliance and the relaxation modulus was conducted using a convenient Laplace transform and an approximated Gamma-function, and a simple but efficient expression to represent the relaxation modulus may be given as
(13)$$E\left(t\right)=\frac{\sigma_{0}}{k\;t^{\beta }}\cdot \frac{\sin\left(\pi \beta \right)}{\pi }\cdot \left(\frac{t_{i}}{\beta }+\frac{1}{t}\right)$$

As mentioned in Ref. [14], relaxation behaviour emphasises short-term processes, while the creep behaviour weights long-lasting processes. Therefore, interconversion between these two groups of responses is frequently desired. Furthermore, to evaluate the viscoelastic damping of polymeric and polymer-based composite materials, it is useful to interconvert the real and imaginary parts of a complex response function within the frequency domain.

The hierarchical structure of the interconversion between viscoelastic properties is displayed in Figure 1.

## 4. Complex Moduli Obtained from EPL Method

A numerical interconversion between the relaxation modulus and the complex modulus is presented below. Laplace and Carson’s transformation and Moivre’s theorem are used to achieve the frequency domain from the time domain.

Using the Laplace transform properties summarised in Table 1, each term of the EPL function (Expression (13)) can be transformed as
(14)$$\bar{f}\left\{e^{\frac{t}{t_{i}}}\right\}=\frac{1}{s+t_{i}}$$
(15)$$\bar{f}\left\{t^{\beta }\right\}=\frac{\beta !}{s^{\beta +1}}$$

After mathematical manipulations, the EPL function of relaxation modulus (Expression (13)), in ℒ-domain, is transformed into
(16)$$\bar{E}\left(s\right)=\frac{\sigma_{0}}{k}\cdot \frac{\sin\left(\pi \beta \right)}{\pi }\cdot s^{\beta }\left[\frac{t_{i}\Gamma \left(1-\beta \right)+s\Gamma \left(1-\beta \right)}{\beta s}\right]$$
where $\bar{E}$ is the rational function of s and Γ is the Gamma function operator. 

Applying Gamma function properties, γ=−β!=Γ(β−1), and rearranging Expression (16) as $E_{\omega }=\left(\sigma_{0}\sin\left(\pi \beta \right)/k\pi \right)\cdot \gamma$, yields
(17)$$\bar{E}\left(s\right)=\frac{E_{\omega }}{\beta }\cdot \left(s^{\beta }t_{i}+s^{\beta +1}\right)$$

Introducing Expression (7) into Expression (17) and after a mathematical manipulation, the complex modulus can be written as
(18)$$E^{*}\left(\omega \right)={\tilde{E}\left(s\right)|}_{s=i\omega }=\frac{E_{\omega }}{\beta }\cdot \left[{\left(i\omega \right)}^{\beta }t_{i}+{\left(i\omega \right)}^{\beta +1}\right]$$

The following known relationship between the two operational functions is obtained
(19)$$E^{*}\left(\omega \right)D^{*}\left(\omega \right)=1$$

From the Theorem of Moivre, Expression (19) can be replaced as
(20)$$E^{*}\left(\omega \right)=E_{\omega }\cdot \frac{\left[e^{iz}\omega^{\beta }t_{i}+ie^{iz}\omega^{\beta +1}\right]}{\beta }$$
where $z=\frac{\pi \beta }{2}$.

Applying the Euler identity and separating even and odd terms, Expression (20) yields
(21)$$E^{*}\left(\omega \right)=E_{\omega }\frac{\omega^{\beta }}{\beta }\;\left[\left(t_{i}\cos z-\omega \sin z\right)+i\left(t_{i}\sin z+\omega \cos z\right)\right]$$

From Expression (21), the real and imaginary parts of the complex modulus are, respectively,
(22)$${ℜ}_{e}=\frac{\omega^{\beta }}{\beta }\left[t_{i}\cos z-\omega \sin z\right]$$
(23)$${ℑ}_{m}=\frac{\omega^{\beta }}{\beta }\left[t_{i}\sin z+\omega \cos z\right]$$

The storage, $E^{\prime }\left(\omega \right)$, and loss modulus, $E^{\prime \prime }\left(\omega \right)$, parts of the dynamic modulus, are, respectively,
(24)$$E^{\prime }\left(\omega \right)=\;E_{\omega }\cdot {ℜ}_{e}=E_{\omega }\cdot \frac{\omega^{\beta }}{\beta }\left[t_{i}\cos z-\omega \sin z\right]$$
(25)$$E^{\prime \prime }\left(\omega \right)=E_{\omega }\cdot {ℑ}_{m}=E_{\omega }\cdot \frac{\omega^{\beta }}{\beta }\left[t_{i}\sin z+\omega \cos z\right]$$

Furthermore, within linear viscoelasticity and from the relationship between the creep compliance and the relaxation modulus, it is possible to rewrite the dynamic modulus as a function of the creep compliance, D(t), and strain, ε(t), rewriting the factor $E_{\omega }$, in Expressions (24) and (25), as follows
(26)$$E_{\omega }=\frac{\varepsilon \left(t\right)\cdot \sin\left(\pi \beta \right)}{D\left(t\right)\cdot k\cdot \pi }$$

Once the creep strain is fitted, it is possible to directly estimate the dynamic behaviour of the polymeric or polymer-based composite material.

## 5. Application of EPL Method

This section presents the application and assessment of the method described above to the spectrum data collected from the literature, namely the studies by Kehrer et al. [15], Berthelot et al. [16], and Ledi et al. [17]. 

### 5.1. Dynamic Modulus Fitted from Kehrer et al. (2018)

Kehrer et al. [15] investigated the dynamic behaviour of a pure polymer (polypropylene—PP). The authors performed dynamic mechanical analysis (DMA) on the pure polymer and also on composite samples (PP reinforced with 30 wt.% of short glass fibres) to experimentally characterise their temperature- and frequency-dependent material behaviour.

The resulting experimental data not only give general information on the material behaviour of the composite subjected to a temperature and frequency load, but also provide input and validation data for the developed material modelling methods.

The DMA tests were conducted in tension mode with different frequencies (0.5 to 50 Hz) and temperature (−50 to 120 °C) loads. In the present study, the test data obtained for the pure PP at 30 °C are used. In preliminary works by Ref. [18], it is shown that for loads corresponding to a strain in the material of ε ≥ 0.60%, a nonlinear material behaviour is initiated. To remain below this value and keep the material within the linear range, the applied static preload and the superimposed dynamic load corresponded to maximum strains of 0.1% and 0.05%, respectively. The contact force was fixed at 5 N.

From the data spectra, the EPL equation fitting was conducted, leading to a maximum difference of ±0.50% between the test data and the fitted curve. The EPL coefficients are shown in Table 2.

Figure 2 shows the evaluation of the storage modulus, $E^{\prime }\left(\omega \right)$, fitted by the EPL equation to the data obtained in the DMA test for the pure PP polymer. The narrow margin of relative error shown in Figure 2b shows that the data adjustment performed by the EPL method presents a very good match to the data points.

From the obtained results, one may note that:
(i)As expected, the calculated instability time, $t_{i}$ (26,178 s), is relatively low, reflecting the typical short lifetime for a pure thermoplastic PP;(ii)The fitting provided by EPL is accurate and may represent the dynamic behaviour of the pure PP thermoplastic polymer;(iii)The fitted curve of the storage modulus exhibits an r-Pearson factor greater than 99%.

The evaluation of the loss factor and the phase angle range is shown in Figure 3. An increase in the phase angle values is shown, which is logical, because the test temperature of 30 °C is higher than the glass transition temperature, $T_{g}$, of PP, which is in the range of −20 °C to 0 °C; however, the test temperature is significantly lower than the softening point of PP, which does not melt below 160 °C.

Comparing the loss factor estimated by the EPL method to the measured loss factor at a temperature of 30 °C and a frequency of 5.0 Hz, the EPL gives η = 0.047, while the measurements show η = 0.050.

### 5.2. Dynamic Modulus Fitted from Berthelot et al. (2008)

Berthelot et al. [16] performed a series of analyses of damping of (i) composite laminates, (ii) composite laminates with interleaved viscoelastic layers, and (iii) sandwich specimens. The composite laminates are constituted of unidirectional E-glass fibre unidirectional mats and silk weave layers (both with 300 g/m^2^) in an epoxy matrix, for a given fibre orientation of 0°, 15°, 30°, 45°, 60°, 75° and 90° with respect to the longitudinal axes of the specimen. The laminate materials with viscoelastic layers are the unidirectional glass fibre composites considered previously in which a single or two viscoelastic layers constituted of Neoprene-based layers (nominal thickness of 0.2 mm) were interleaved. The sandwich specimens were constructed with the [0/90]s cross-ply laminates as skins, enclosing 15 mm thick PVC closed-cell foams with three different densities (60, 80 and 200 kg/m^3^).

The composite laminates were fabricated by hand lay-up in plates of different dimensions; these were cured at room temperature with pressure, using a vacuum moulding process, and then post-cured in an oven. The plates were fabricated with eight layers in such a way to obtain the same plate thickness (nominal value of 2.4 mm) with the same fibre volume fraction (nominal value of 0.40). 

The experimental investigation of the damping of the different materials involved using beam test specimens and an impulse technique. Each test specimen was supported horizontally as a cantilever beam in a clamping block. An impulse hammer was used to induce the excitation of the flexural vibrations of the beam, and the beam response was detected using a laser vibrometer.

The evaluation of damping was performed on beams of different lengths (160 mm, 180 mm and 200 mm) to result in a variation of the values of the peak frequencies. Beams had a nominal width of 20 mm and a nominal thickness of 2.5 mm. The results were reported for the first three bending modes. 

The following two assumptions were made about this investigation:
The test temperature, not reported in the paper, is assumed to be around 25 °C;The loss factor is β and $t_{i}$ dependenton the EPL method, therefore the coefficient k is assumed to be the unit for all samples and fibre orientation.

From the loss factor measured for 0°, 45°, and 90° of fibre orientation, the EPL equation fitting was conducted applying the least-squares method to determine the best fit curve to test data. The average relative error to the three adjustments was 4.43%. The EPL coefficients are shown in Table 3. 

From Table 3, for given fibre orientation from the longitudinal axis of the specimen, it is observed that the value of β and $t_{i}$ coefficients are fibre orientation dependent. Figure 4 shows the evaluation of the loss factor, η, fitted by the EPL equation, regarding the data of the impulse technique test for unidirectional glass fibre composites. It is observed that the damping for specimens where the fibre orientation is transverse to their longitudinal axis (45° and 90°) is greater than for specimens where the fibre orientation is parallel to its the longitudinal axis. 

### 5.3. Dynamic Modulus Fitted from Ledi et al. (2018)

Ledi et al. [17] present an identification method of the viscoelastic material properties (shear modulus and loss factor) of a symmetric three-layered viscoelastic sandwich beam. The authors performed experimental vibration tests to determine resonant frequencies and loss factors for different bending modes.

The viscoelastic sandwich beam tested by the authors was composed of three layers, bonded together: two aluminium elastic face sheets and one viscoelastic core layer, made of polyurethane dielectric resin (made of polyol and isocyanate). Two types of sandwich beams (beam 1 and beam 2) distinguished by their respective core and faces thicknesses were considered. The geometrical and material properties of the sandwich beams are reported in Table 4.

The experimental setup to measure sandwich beam frequency response curves comprised of an eloetrodynamic shaker, a laser vibrometer and a sandwich beam sample. The shaker generated vibrations that were controlled and measured by a controller device. The frequency excitation produced by the shaker ranged from 4 Hz and 1500 Hz. The vibration amplitude of the shaker was controlled by an accelerometer. The amplitude of the shaker was chosen to minimise nonlinear vibrations (typically 0.02 mm of displacement between 5 Hz and 157.5 Hz, and a 1 g acceleration between 157.5 Hz and 1500 Hz). A laser vibrometer was used to measure the amplitude of the beam’s vibration; this equipment was pointed at the free extremity (tip) of the beam (cantilever), whose boundary condition was clamped-free. The test apparatus used in the experiments allowed adjustments of the clamping length—four lengths (from 318 to 475 mm) were tested.

For a linear, homogeneous and isotropic material, the EPL function of storage modulus Expression (24) may provide the definition of the storage shear modulus, $G^{\prime }\left(\omega \right)$, whose relationship may be established by the complex Poisson’s ratio, $\upsilon^{*}\left(\omega \right)$, as
(27)$$E^{\prime }\left(\omega \right)=2G^{\prime }\left(\omega \right)\cdot \left[1+\upsilon^{*}\left(\omega \right)\right]$$

Given the reduced dependence on the frequency of Poisson’s ratio, its value can be considered a real and constant quantity [19]. 

From the shear modulus measured, the EPL function of the *shear storage modulus* was fitted and the obtained coefficients are presented in Table 5. 

The evaluation of the in-plane shear storage modulus for both beams is shown in Figure 5, where it can be noted that the fitting is very close to the data points, providing a good description of the behaviour of the identified shear modulus quantities over a wide range of frequencies. It should be noted that because the adjusted curve refers to the spectrum of two distinct data tests, the evaluation of the relative error through the dispersion plot was not used in this adjustment; as an alternative, the r-Pearson coefficient was used to evaluate the quality of the fitting: values of 0.962 and 0.981 were obtained for beams 1 and 2, respectively.

## 6. Conclusions

This paper presents a new approximate interconversion scheme to convert the measured dynamic moduli into creep compliance modulus and relaxation modulus. The proposed method uses a convenient Laplace and Carson transform and its relationship to the Fourier Domain, coupled with the application of the Moivre and Euler Theorem.

The EPL creep fitting method described in Ref. [1] has been converted to the EPL dynamic modulus equation and the data provided in the literature have been successfully fitted. In particular, the test data concerning three different polymeric or polymer-based composites were used to validate the numerical approach: a pure polymer thermoplastic resin specimen, an E-glass-epoxy composite laminate, and an elastic-viscoelastic sandwich beam.

The results obtained show the accuracy of the EPL Method in fitting the data from creep, relaxation, and dynamic tests. Moreover, the method can show the three stages of creep (long-time behaviour). In addition, it is shown that fitting based on the EPL method is easy to implement since only three terms need to be found in any data spectrum, and no pre-smoothing procedure is required to ensure a good fit. Furthermore, the stability of this method is demonstrated by the achieved Pearson correlation coefficients, which was higher than 0.96 for the different examples.

## Figures and Tables

**Figure 1 materials-13-05213-f001:**
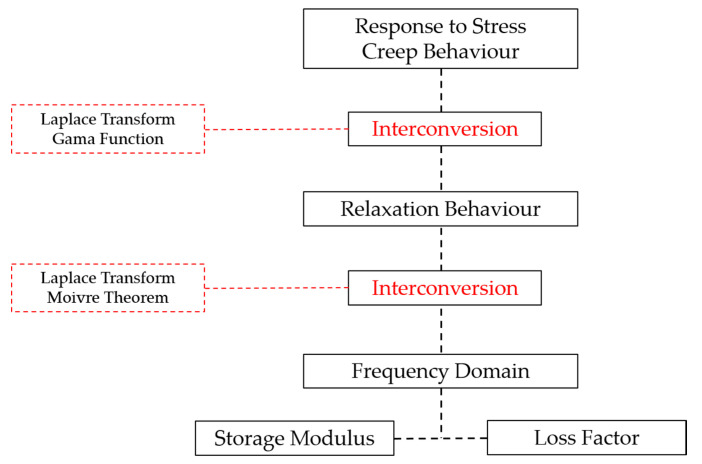
Hierarchy of interconversion relations (adapted from Ref. [14]).

**Figure 2 materials-13-05213-f002:**
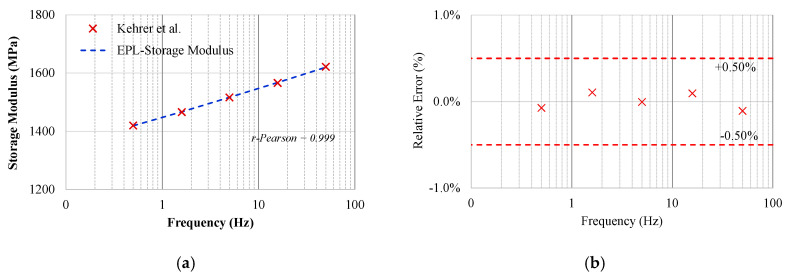
Storage Modulus for polypropylene (PP): (**a**) EPL fitting and (**b**) Relative Error in %.

**Figure 3 materials-13-05213-f003:**
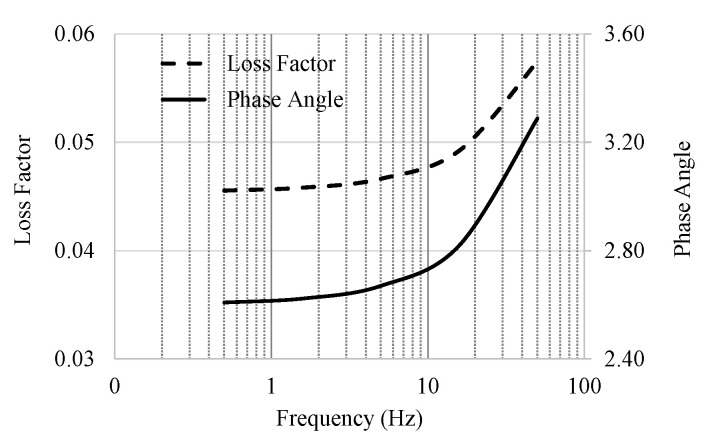
Loss Factor and Phase Angle of pure PP from EPL frequency-domain fitting.

**Figure 4 materials-13-05213-f004:**
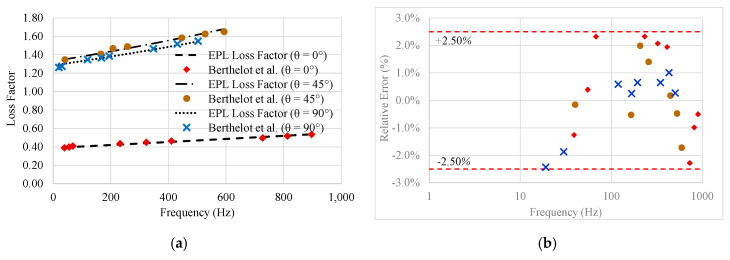
(**a**) Loss Factor as a function of the frequency for different fibre orientations for unidirectional glass fibre composites and (**b**) Dispersion of the Relative Error in %.

**Figure 5 materials-13-05213-f005:**
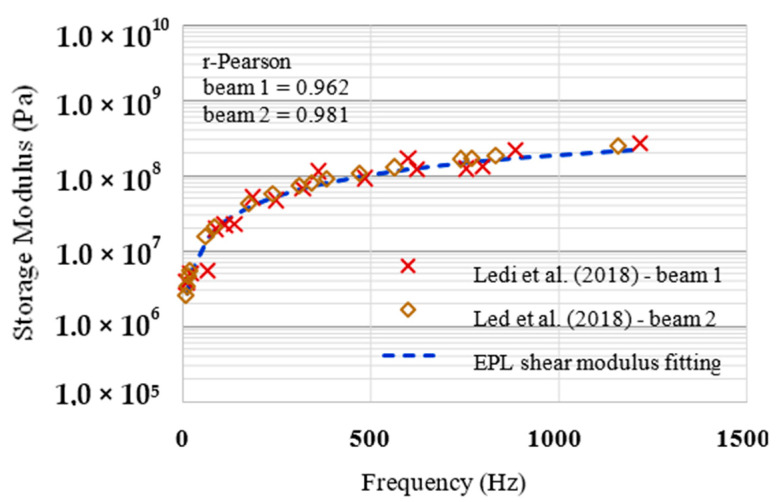
In-plane Shear Storage Modulus for sandwich beams.

**Table 1 materials-13-05213-t001:** Laplace transform properties.

f(t)	$\hat{f}\left(t\right)$
$e^{\alpha t}$	$\frac{1}{s-\alpha }$
$t^{n},\;n>-1$	$\frac{n!}{s^{n+1}}$

**Table 2 materials-13-05213-t002:** Parameters of the Exponential-Power Law (EPL) equation for Dynamic Moduli.

$\sigma_{o}(\mathrm{MPa})$	k	β	$t_{i}(s)$
3.00	58.08	0.0289	26,178

**Table 3 materials-13-05213-t003:** Parameters of EPL equation for Loss Factor.

θ°	k	β	$t_{i}(s)$
0°	1	0.235	45,707
45°	1	0.589	34,443
90°	1	0.579	36,275

**Table 4 materials-13-05213-t004:** Material properties and dimensions for beam 1 and beam 2.

**Elastic Layers**	
Material	Aluminium
Young’s Modulus	$E_{f}=$6.9 × 10^10^ Pa
Poisson’s ratio	$v_{f}=$0.3
Density	$\rho_{f}=\;$2766 kg/m^3^
Thickness	$h_{f,1}=\;$1 mm and $h_{f,2}=\;$0.5 mm
**Viscoelastic Layer**	
Material	Polyurethane dielectric resin
Poisson’s ratio	$v_{f}=$0.3
Density	$\rho_{f}=\;$1550 kg/m^3^
Thickness	$h_{c,1}=\;$1 mm and $h_{c,2}=\;$2 mm
**Whole Beam**	
Length	L = 500 mm
Width	b = 30 mm
Thickness	h = 3 mm

**Table 5 materials-13-05213-t005:** Parameters of the EPL equation for the in-Plane Shear Storage Modulus.

k	β	$t_{i}(s)$
0.68	0.98	2469.136

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
