# Peer review of "A New Viscoelasticity Dynamic Fitting Method Applied for Polymeric and Polymer-Based Composite Materials"

_materials, 2020, doi:10.3390/ma13225213_

Round 1

Reviewer 1 Report

The paper is well structured and written. The proposed EPL method seems to be convincing.

In the context of viscoelectic material, experimental quantification of uncertainties is often significant. Thus, stochastic modelling is needed. The authors use the reference [16]. In this research team, some works deal with the non deterministic analysis. I think this point should be relevant in the state of the art of this paper.

There is a problem with equations 23 and 24, perhaps it is linked with the pdf file.

In Figure 2, the results of Kehrer are given per point. So why the relative error is curve? It seems to be linear between two points?

Is it possible also to quantify the EPL error for Berthelot et al ?

Is it possible also to quantify the EPL error for Ledi et al. ?

Author Response

The authors thank the reviewer for his positive assessment of our work. In the attached, please find a detailed response to the specific remarks and the actions prompted by each remark.

Reviewer 2 Report

The author's approach to the processing and analysis of dynamic mechanical data of materials obtained on the basis of polymers proposed by the authors of the article is important from a scientific point of view. The authors sought to obtain a dynamic expression of the modules from the exponential power law (EPL) method of creep susceptibility and relaxation module functions, using the Carson and Laplace transform functions and their relationship to the Fourier transform and Moivre theorem. They verified this method on the basis of data from the literature (3 scientific articles), it is a pity that not on their own experimental data.
The concept of the reliability of the described method appears in the conclusions section. Are you sure you can draw this type of conclusions? Shouldn't the model be validated on a larger number of data (cases)?
The way in which the authors describe the literature data (items [14-16]) is quite brief. Maybe it would be worth presenting this data more broadly so that readers do not have to check them in source articles?
I have doubts whether it is permissible for the Authors to refer to another article of their own, which is still under review and has not been published (item [8]). At this point, the reviewer is not able to verify this data because he has no access to it.
The article contains 18 literature items, half of which are more than 20 years old.
The formulas (23) and (24) in the reviewed version coincide and are illegible - perhaps this problem appeared when saving the article to pdf format.
In my opinion, the Conclusion section should be rebuilt, it should be more concise, without descriptions of what was presented in the Introduction and without referring to the reading.

Author Response

The authors thank the reviewer for his positive assessment of our work. In the attached document, please find a detailed response to the specific remarks and the actions prompted by each remark.

Reviewer 3 Report

This paper reports on an analytical methodology for converting between static creep/relaxation and dynamic viscoelastic testing, which was demonstrated on several polymeric materials by comparing to experimental data in the literature, showing good agreement. The methodology and results appear valid, and no significant grammatical or formatting errors. This paper is recommended for publication.

Author Response

The authors welcome the Reviewer’s positive comments about the paper.

Round 2

Reviewer 2 Report

Thank you very much for the detailed and substantive answers to all comments. I maintain the opinion that the article deals with an important topic. The authors made corrections, so I believe the article may be published.